# Unusual Presentation of Clear Cell Odontogenic Carcinoma: Case Report and Literature Review

**DOI:** 10.3390/diagnostics12041019

**Published:** 2022-04-18

**Authors:** Raúl Verdeja, Edouard Stauffer, Tommaso Lombardi

**Affiliations:** 1Department of Surgery, Service of Maxillofacial and Oral Surgery, University Hospital and Faculty of Medicine, 1205 Geneva, Switzerland; 2Private Pathology Institute (Promed), 1700 Fribourg, Switzerland; edouard.stauffer@promed-lab.ch; 3Unit Oral Medicine and Maxillo-Facial Pathology, Department of Surgery, Service of Maxillofacial and Oral Surgery, University Hospital and Faculty of Medicine, 1205 Geneva, Switzerland; tommaso.lombardi@unige.ch

**Keywords:** odontogenic tumor, clear cell odontogenic carcinoma, histological and immunohistological diagnostic, asymptomatic, surgical treatment

## Abstract

Clear cell odontogenic carcinoma (CCOC) is a rare, aggressive epithelial neoplasm of the jaw first described in 1985 by Hansen and classified as an odontogenic malignant tumor in the 2005 WHO classification. To date, only 117 cases have been reported in the literature written in English. In this paper, we report the atypical presentation of a pericoronal localized tumor in the right mandibular wisdom tooth discovered during a routine radiologic examination. The patient, a 44-year-old healthy female, was referred by her general dental practitioner for examination of temporomandibular dysfunction and recurrent myofascial pain. Anamnesis and clinical examination did not suggest tumoral disease. The osteolytic lesion was removed, and histological examination revealed a clear cell odontogenic tumor (CCOT) of the right posterior lower jaw. Segmental lower jaw resection was performed, and a free iliac crest bone transplant was fixed with mandibular reconstruction plate. No cervical neck dissection was needed. A five-year follow-up examination shows excellent local and systemic recovery and no sign of tumor recurrence. The patient still suffers from bruxism and myofascial related pain, which are treated conservatively with a Michigan splint and physiotherapy. Based on this additional case, we review the literature and discuss the challenging diagnostic aspects, the unusual clinical presentation, and the treatment of CCOC.

## 1. Introduction

Clear cell odontogenic carcinoma (CCOC) is a rare intraosseous jaw tumor, first described in 1985 by Hansen [1]. Due to its malignant characteristics, CCOC has been considered a malignant tumor in the WHO classification since 2005 [2]. Until its classification as an odontogenic tumor by the WHO [3] in 1992, it was often designated as clear cell ameloblastoma [4]. Representing only 6.1% of all odontogenic tumors, it occurs most frequently in females in their 5th to 7th decades, and the mandible is the most frequent location. Clinical signs include painless swelling, loosening of teeth, and paresthesia. Sixty percent of cases show soft tissue involvement in the anterior part of the mandible. The radiology is characterized by uni- or multilocular radiolucency with bone destruction. A neighboring tooth can show divergence in tumor volume, as well as root resorption in some cases [5].

The tumor is formed from lobules of cells with only clear cytoplasm, or mixed with cells containing eosinophilic cytoplasm that are rich in glycogen, and separated by bands of collagen.

Histopathological diagnosis requires special examinations to differentiate CCOC from other clear cell lesions. Specifically, special stains (Mucicarmine, Congo red, Periodic acid-Schiff) or immunohistochemical staining (Cytokeratin, Epithelial membrane antigen, S-100 protein, Vimentin) are needed. The expression of cytokeratin-19 and the epithelial membrane antigen are determinants for diagnosis, as well as negativity for Vimentin, S-100 protein, and smooth muscle actin. Harbhajanka et al. presented a challenging diagnosis based on fine needle aspiration [6]. CCOC shows translocation of Ewing sarcoma breakpoint region 1 gen RNA-binding protein 1 (EWSR1)-activating transcription factor (ATF1). These techniques seem to be very helpful in distinguishing metastatic and odontogenic tumors [7]. In a recent review of the literature, Desai et al. [8] hypothesize that CCOC is the intraosseous counterpart of clear cell carcinoma (CCC) of the salivary glands.

## 2. Case Report

We report the unusual presentation of clear cell odontogenic carcinoma in a 44-year-old woman. She was referred by her general dental practitioner with temporomandibular complaints such as typical bilateral myofascial pain syndrome with bruxism. Clinical examination and radiographic investigation with an OPG revealed a deeply impacted right mandibular third molar (48), showing a 3 cm radiolucent lesion located above the crown (Figure 1a). A cystic odontogenic lesion of the jaw related to the impacted wisdom teeth was shown. Radiological examination was completed with Cone-Beam CT to obtain more information concerning the localization of the inferior alveolar canal. However, the radiologic aspects of the osteolytic lesion remained non-pathognomonic. The patient had no clinical symptoms related to the tumor.

Under general anesthesia, the osteolytic lesion was treated by surgical excision, and the tooth was extracted. The excised material measured approximately 2 cm and was firm in consistency. This first surgical procedure did not cause any lesion of the alveolar nerve.

Microscopic examination revealed a proliferation of islands of clear cells presenting in places a peripheral basaloid cell component, separated by more or less hyalinized fibrous tissue (Figure 2a–c), with nuclear atypias in some areas (Figure 3b), but mitoses were sparse (Ki67 < 1%). This cellular mass infiltrated most of the resected material. Clear cells were PAS-positive and became negative following diastase digestion (Figure 3a). Infiltration of vascular and perineural structures was not found in this sample of material. Further examination showed positivity for anti-cytokeratin antibodies (CK18, CK19 and MNF116) (Figure 4a–c), CK-HMW (not shown), anti-EMA antibodies (Figure 4d), and P 53 (Figure 4e). Negative reaction was observed with antibodies against hormonal receptors (ER, PgR), GCDRP15 (not shown), HMB45 (not shown), Alcian Blue (Figure 5a). and CD10 (Figure 5b).

The patient was then referred to University Hospital for tumor staging and treatment. No metastatic lymphatic nodes or distant metastasis were detected. The definitive treatment was performed by cervicotomy for segmental resection of the right mandibular angle region and reconstruction of the 5 cm bone defect with a free bone transplant taken from the right iliac crest and osteosynthesis plate; no neck dissection was needed. Postoperative recovery was uneventful, and clinical follow-up after 5 years showed no sign of recurrence. Aesthetically and functionally, she is similar to the preoperative situation. Perfect osseous healing can be noticed on the follow-up CBCT 1 and 3 years after the major operation (Figure 1b). She became accustomed to the anesthesia of the third right trigeminal branch and had no complaints about it.

The five-year checkup took place with clinical examination and injected conventional CT. No sign of tumor recurrence could be highlighted. She still suffers from bruxism and bilateral myofascial pain; conservative treatment took place after recovery from the second operation, with a Michigan splint and physiotherapy.

## 3. Discussion

Clear cell odontogenic tumors (CCOT) were defined by the World Health Organization (WHO) in 1992 as benign tumors with local invasive behavior [9]. The malignant potential of such lesions was recognized in the following years, after articles were published reporting metastasis and deaths associated with CCOTs [2]. In 2005, the WHO confirmed the classification of CCOT as a malignant tumor, and CCOC was definitively adopted as a new nomenclature for the tumor [10]. Knowledge of this tumor is based on a limited number of case reports. After Peraza Labrador et al.’s [11] systematic review, there have been 117 published cases in the English-language literature. Ninety-four were included in the systematic review from Loyola et al. [12], and two further ones have been published since 2015 [4,13], in additional to our case in question. The publications reveal that CCOCs tend to entail 4 cm large lesions of the posterior mandible in women in their sixth decade. Less-frequent symptoms include slow-growing swelling over a long rate of development, producing bone resorption and soft tissue invasion.

Radiographic findings are similar to those associated with other intraosseous lytic lesions. The literature does not emphasize the fact that CCOC can mimic benign or more aggressive osteolytic jaw tumors, nor does it describe their metastasis process.

Clinical signs and radiology of CCOC show low specificity. Diagnosis is equally difficult using conventional histopathology, with clear cells being characteristic but not pathogno-monic of this lesion. Special stains, such as mucicarmine, Congo Red and Periodic acid-Schiff (PAS), or immunohistochemical staining, such as cytokeratin (CK), epithelial membrane antigen (EMA), S-100 protein, and Vimentin, are used to establish the diagnosis. Expression of CK-19 and EMA is a consistent finding, and the tumor also shows negativity for Vimentin, S-100 protein, Desmin and smooth muscle Actin [6]. Differential diagnosis should include odontogenic and non-odontogenic lesions with clear cell change, such as calcifying epithelial odontogenic tumors, clear cell ameloblastoma or intraosseous salivary gland tumors (epithelial-myoepithelial carcinoma) or metastatic tumors (clear cell renal carcinoma, melanoma, salivary gland or mammary carcinoma).

The literature update (Table 1) based on systematic reviews [11,12] includes symptoms such as swelling, pain, and tooth mobility. CCOC is a tumor of middle-aged patients, arising in females 65% of the time and in males 35% of the time, with predominance in the lower jaw. Radiological findings show a radiolucent lesion that can mimic other bony lesions of the jaws. Treatment is a wide surgical resection without neck dissection and reconstruction in 95% of the cases. Adjuvant treatment with radiotherapy was only performed in 32% of cases.

Recently, it has been published that CCOC is almost identical to CCC in terms of histomorphological, immunohistochemical, and molecular properties [8], and hence CCOC can be defined as the intra-osseous variant of CCC.

Compared to the published cases, ours was an incidental finding from a radiological examination for temporomandibular dysfunction. The initial treatment, based on the false diagnosis of an impacted tooth with pericoronal cystic lesion without any histological examination, could have been avoided. Histological and immunohistological studies seem to be the only sure method that pathologists can use for this challenging tumor. As numerous cases of malignant development have been described, preoperative tumor staging is needed. In our case, it was possible to perform minimal resection of the jaw with free bone reconstruction. Clinical and radiological follow-up is, of course, needed. A crossover evaluation of the 117 cases reported in the English-language literature can be seen in Table 1.

## 4. Conclusions

A review of the literature shows the versatility and difficulty of diagnosing CCOC [14]. As with every malignant tumor, preoperative staging with clinical, radiological, and histopathological assessment is needed. Our literature update confirms the presence of symptoms such as swelling, pain, and tooth mobility. CCOC is a tumor of middle-aged patients, arising in females 65% of the time and males 35% of the time, with predominant localization in the lower jaw. The reported radiological findings are of a radiolucent lesion that could mimic other bony lesions of the jaws. Treatment is a wide surgical resection without neck dissection and reconstruction in 95% of the cases. Adjuvant treatment with radiotherapy was only performed in 32% of cases.

Our case is a perfect example of the variability of CCOC, with no clinical signs and with radiological findings suggesting aggressive growth. Only histological and immunohistological studies were able to establish the final diagnosis. Follow-up after 5 years showed no recurrence and minimal functional impairment.

In the latest World Health Organization (WHO) tumor classification from 2017 [15], CCOC remains unchanged from the 2005 version.

## Figures and Tables

**Figure 1 diagnostics-12-01019-f001:**
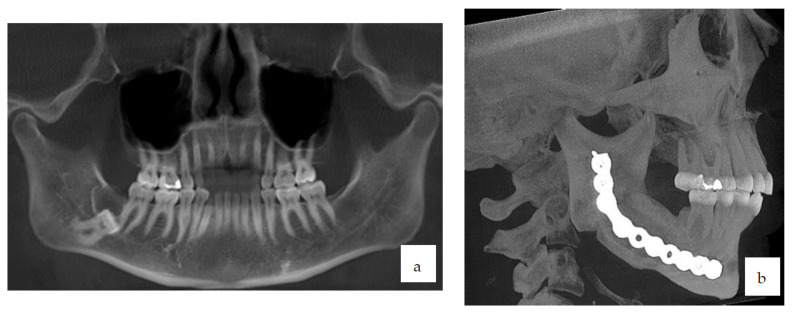
(**a**) Preoperative panoramic radiography; (**b**) cone beam computed tomography 3 years after surgery.

**Figure 2 diagnostics-12-01019-f002:**
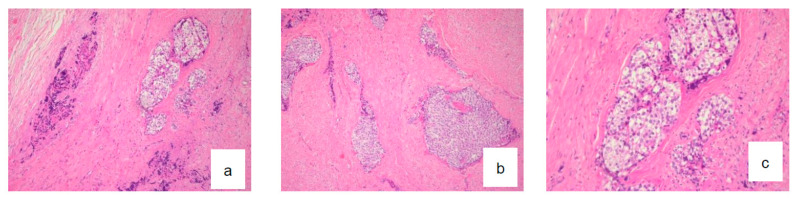
Histological aspect of CCOC on hematoxylin-eosin stain (**a**) ×4, (**b**) ×10, and (**c**) ×20.

**Figure 3 diagnostics-12-01019-f003:**
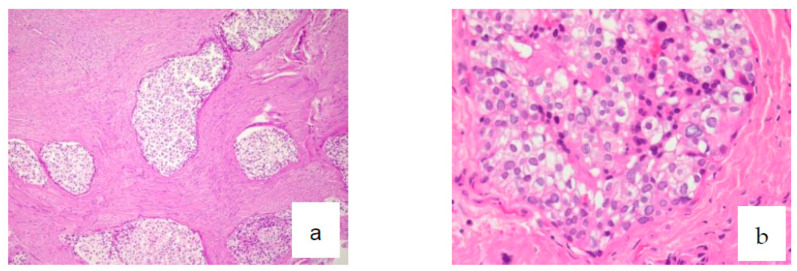
(**a**) PAS positivity for glycogen (×10), (**b**) nuclear atypias (Ki67 < 1%) (×20).

**Figure 4 diagnostics-12-01019-f004:**
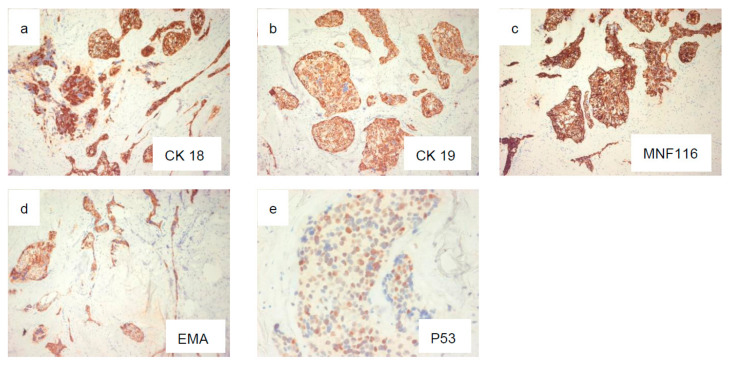
Positivity for AC anti-CK18 ((**a**), ×4), CK19 ((**b**), ×10), MNF116 ((**c**), ×10), EMA ((**d**), ×4), and P53 ((**e**), ×40).

**Figure 5 diagnostics-12-01019-f005:**
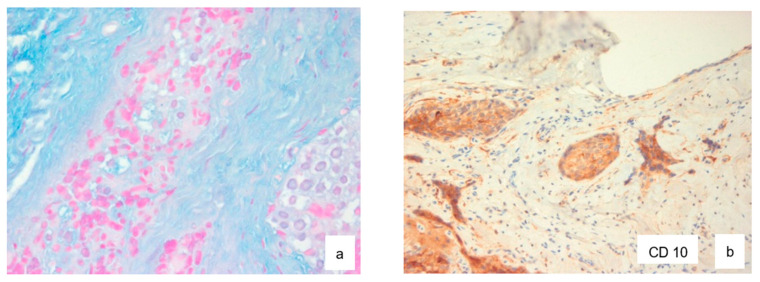
Alcian Blue staining ((**a**), ×40) and CD 10 immunohistochemistry ((**b**), ×10).

**Table 1 diagnostics-12-01019-t001:** Up-to-date review of the English-language literature with 117 CCOC cases, (x = number of cases, n = total published) [5,11,12,13].

		(x/n)	Percent
Cases	117		
Age	Mean 55.4 (17–89 years old)		
Gender	Female	76/117	64.9%
	Male	41/117	35.0%
Size	Mean 4.36 cm (range 3–10 cm)		
Location	Lower jaw	96/117	82.05%
	Upper jaw	20/117	17.95%
	Most commonly mandible body	42/117	35.90%
Signs and symptoms	Swelling	74/117	80.40%
	Pain	31/117	41.30%
	Tooth mobility	23/79	29.11%
None	1/79	1.27%	
Histologic pattern	Ameloblastic	13/79	16.45%
	Biphasic	62/79	78.50%
	Monophasic	4/79	5.06%
Surgical margins	Positive	17/41	41.50%
	Negative	24/41	58.54%
Recurrence	Yes	44/84	52.40%
	No	40/84	47.60%
Local recurrence	Present	33/81	40.74%
	Absent	44/81	54.32%
Metastasis	Distant metastasis	20/117	17.10%
	Lung	17/117	14.50%
	Regional metastasis	10/117	8.55%
	Neck	10/117	8.55%
No reported metastasis		87/117	74.37
Treatment modalities	Local resection	64/102	62.75%
	Curettage or enucleation	5/102	4.90%
	Hemimandibulectomy	23/102	22.55%
	Hemimaxillectomy	10/102	9.80%
Adjuvant therapy	Radiotherapy	21/81	25.93%
	Chemotherapy	5/81	6.17%
Follow up (88 patients)	Mean	26.10 months	
	Range	1 to 152 months	
Outcome	Healed from CCOC	82/94	87.20%
	Dead from CCOC	12/94	12.80%

## Data Availability

Not applicable.

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
