# Peer review of "Unusual Presentation of Clear Cell Odontogenic Carcinoma: Case Report and Literature Review"

_diagnostics, 2022, doi:10.3390/diagnostics12041019_

Round 1

Reviewer 1 Report

The authors report a case of clear cell odontogenic carcinoma and perform a review of published cases. Here are some suggestions and comments:

Abstract:

1) The abstract should not have any references.

2) What do authors mean by “Asymptomatic clear cell odontogenic tumors of the right posterior lower jaw could be diagnosticated” ………. lines19-20

Introduction:

1) Reference 2 is missing from this section. References should appear in the order they are cited.

2) Line 52, citation 7……….the authors cited the wrong reference.

Case report:

  • Was CBCT imaging done before surgery? If yes, it may be nice to include it.
  • Did the patient have any paresthesia from the tumor or the impacted tooth at time of diagnosis? This should be reported.
  • Was the inferior alveolar nerve comprised with the surgical procedure?
  • Figures are not appropriately labeled.

(a) In microscopic examination, authors refer to Fig 4b for Alcian Blue and Ki67, whereas Alcian Blue is Fig 5 and Ki67 is Fig 3.

(b) Label Fig 3 as A (for PAS) and B (for Ki-67, not Nuclear atypies)

(c) Similarly, label Fig 4 as A-E, and Fig 5 as A and B.

  • Lines 77-78………………..Examination showed positivity for S100 antibodies, and refer to Fig 4 which does not show any S100 labeling. Additionally, on lines 50-51, authors write that negativity for vimentin, S-100 and SMA are diagnostic for CCOC.
  • Lines 79-80………….Negative reaction could be demonstrated……CD10 (Fig 5)…….Fig 5. This figure shows only CD10 and ER, PR, GCDRP15, HMB 45 are not shown.
  • Line 89-90……..Perfet osseous healing…..second operation. What was the first operation?

Discussion:

  • Lines 118-120……The malignant potential….CCOT. Cite this sentence.
  • Lines 132-133….Poorly described in the lietrature is the way. Please rephrase this sentence.
  • Table 1……provides a summary of demographic, clinical, radiographic, and treatment data of the 117 published cases of CCOT. But no where do authors discuss these features of CCOT except for a line or two in the Conclusion. These should be added to the discussion section.

Authors should include a methodology section as to how the search For previously published CCOT cases was performed. What words and platforms were used for the search?

Author Response

Dear Reviewer 1

Thank you for your extensive review. I will answer following your chronology

Abstract:

1- I removed the references

2- Lines 19-20:  We discovered incidentaly this osteolytic lesion by routine radiologic examination allthough the patient came for other problems specifically bruxism with bilateral muscular and TMJ symptoms.

Introduction:

1- The reference numbers have to be modified regarding to the modification in the Abstract of course.

2- You are right it is Nr 6

Case report:

A CBCT was made preoperatively, We prefered to show the first image by panoramic radiograph as most of colleagues do not dispose of CBCT device. But of course we could replace it by CBCT image.

The patient had no symptoms related to the tumor. We will add a sentence to clarify the absence of neurological signs.

The first surgery produced no lesion of the alveolar nerve. But unfortunately the second procedure with segmental resection of the lower jaw provided a definitiv anesthesia. As we comment at the end of the case report, this young woman never complained about the anesthesia.

We will correct the unappropiated labelling of the of the figures. Sorry for that.

Line 77-78 Will be corrected. Off course the negativity for S100 is correct.

Line  79-80 We will add that only CD10 is schown on the figure.

Line 89-90 The first operation has been: osteotomy, surgical extraction oft 48, tumorectomy and preservating osteosynthesis.

Discussion:

118-120 Reference will be added

132-133 The sentence will be corrected

Table 1 A discussion of the table will be added

Corrected version will be sent in the next 2 days.

Thanks once more, best regards

R. Verdeja

Reviewer 2 Report

The manuscript is sufficiently clear, with adequate references to the literature and it is presented in a well-structured manner. The article aims to present an unusual case of CCOC with no clinical signs and radiological findings suggesting aggressive growth. However, these data are not clinically significant to justify the scientific interest. They can be regarded as occasional finds. The absence of symptoms and the radiological interpretation of this case report are variable without any impact on patient management: they do not help clinicians to improve chances of correct diagnosis or to decide on other surgical procedures. In fact, even the authors of the article treated the case according to the usual methods described in the literature on osteolytic lesions: first it was treated by surgical excision with tooth extraction then, only after microscopic examination and immunohistochemical staining, segmental resection of the mandible.

Author Response

Dear Reviewer 2,

Thanks you for the very helpfull evaluation of our paper.

Interesting in this case is the absence of clinical signs and the unusual radiological aspect. These makes a difference to most publicated cases in the literature. According to the symtoms and radiology we suspected a follicular dentogenic cyst of impacted tooth 48. Following this first intuition, I agree with you, we probably did an operation to much. A biopsy in local anesthesia could have brought us to the chalanging diagnostic.

The interest of our paper is to show the variability of clinical presentation and the necessity to include this seldom kind of tumor in the differencial diagnostic of assymptomatic osteolytic lesion.

Second we tried to focus our publication on the challenging diagnostic and the local treatment of Clear Cell Odontogenic Carcinoma correlated with a review of the literature.

Best regards

R. Verdeja

Reviewer 3 Report

The manuscript is well-written and scientifically sound. In the introduction, please separate the section for the aim and objective of the current study. 

In figures 2 to 5, please add the scale bars and also mention them in the figure legends. I would suggest explaining the findings briefly in the figure legends.

Author Response

Dear Reviewer 3

I will follow your recommandations. I added the aim and objective in the introduction. I will also adapt the figures 2 to 5 after your instructions.

The grammatical language will be professional corrected.

Best regards

R. Verdeja